# Animal Model for Glucocorticoid Induced Osteoporosis: A Systematic Review from 2011 to 2021

**DOI:** 10.3390/ijms23010377

**Published:** 2021-12-29

**Authors:** Andy Xavier, Hechmi Toumi, Eric Lespessailles

**Affiliations:** 1EA 4708 I3MTO Laboratory, Orleans University, 45067 Orleans, France; andy.xavier@univ-orleans.fr (A.X.); hechmi.toumi@univ-orleans.fr (H.T.); 2Translational Medicine Research Platform, PRIMMO, Regional Hospital of Orleans, 45007 Orleans, France; 3Department Rheumatology, Regional Hospital of Orleans, 14 Avenue de L’Hopital, 45007 Orleans, France

**Keywords:** glucocorticoid induced osteoporosis, bone, animal models, bone loss, therapy, methylprednisolone, dexamethasone

## Abstract

Clinical and experimental data have shown that prolonged exposure to GCs leads to bone loss and increases fracture risk. Special attention has been given to existing emerging drugs that can prevent and treat glucocorticoid-induced osteoporosis GIOP. However, there is no consensus about the most relevant animal model treatments on GIOP. In this systematic review, we aimed to examine animal models of GIOP centering on study design, drug dose, timing and size of the experimental groups, allocation concealment, and outcome measures. The present review was written according to the PRISMA 2020 statement. Literature searches were performed in the PubMed electronic database via Mesh with the publication date set between April, 2011, and February 2021. A total of 284 full-text articles were screened and 53 were analyzed. The most common animal species used to model GIOP were rats (66%) and mice (32%). In mice studies, males (58%) were preferred and genetically modified animals accounted for 28%. Our work calls for a standardization of the establishment of the GIOP animal model with better precision for model selection. A described reporting design, conduction, and selection of outcome measures are recommended.

## 1. Introduction

Despite their side effects, glucocorticoids (GCs) continue to be prescribed in many diseases because of their immunomodulatory capacities. However, therapy with long-term GCs leads to deleterious effects due to their systemic impact on the metabolism including cardiovascular, endocrine, dermatologic, muscular, and skeletal effects including bone fragility and aseptic osteonecrosis [1].

Glucocorticoid induced osteoporosis (GIOP) is the leading cause of secondary osteoporosis. GC intake is associated with large morbidity and increased mortality [2]. In patients with chronic corticosteroid therapy, the annual incidence rate of vertebral fractures was 3.2% (95% CI: 1.8–5) and 5.1% (95% CI: 2.8–8.2) in patients initiating treatment [3]. However, an increased risk of fragility fracture may be observed within the first three months of treatment [3,4]. Fragility fractures are the most common serious adverse events related to GIOP. Vertebral fractures, most often asymptomatic, may occur soon after exposure to GCs when bone mineral density (BMD) is rapidly decreasing [5]. Vertebral fractures are particularly associated with GIOP, although the risk of hip fractures is also increased [2,4].

Bone loss preferentially affects trabecular bone rather than cortical bone [6]. GCs can cause localized alterations in bone microarchitecture, resulting in micro-lesions that decrease bone strength. These localized alterations of microarchitecture have been shown to be correlated with GC intake [7]. These unique effects explain why GC exposure is associated with an increased risk of fracture at higher BMD values than in postmenopausal osteoporosis. The decrease in bone strength associated with GIOP seems to be rapidly reversible, as clinical observations showed that the prevalence of fractures decreased from the third month at the end of treatment [5,8].

GCs alter the formation/resorption balance leading preferentially inhibition of bone formation [4,7]. During the first few months of treatment with GCs, the loss of bone density is greatest because GCs not only inhibit bone formation, but also accelerate bone resorption [6,9]. This transient increase in bone resorption has been shown to be in part related to treatment with GCs but also to the underlying inflammatory disease and it has been clearly demonstrated that inflammation promotes osteoclastic differentiation [10,11].

To further explore GIOP, it is of utmost importance to use animal models that show similarity to human pathophysiology, in order to carry out efficient preclinical studies and to test new compounds. Currently, the drugs used to treat GIOP are most often those used for postmenopausal osteoporosis (bisphosphonates, SERMs, and parathyroid hormone derivatives) [12]. However, these treatments mainly affect bone metabolism and do not correct other side effects of GIOP such as muscle wasting. These drugs are considered as active comparators in animal studies aiming at the development of new molecules. However, for preclinical studies, it remains difficult to find a suitable animal model that mimics human skeletal development. The use of large human-like animals such as non-human primates have been preferred due to their similarity concerning reproductive, anatomical, and physiological characteristics [13]. However, the use of these large animals such as dogs, pigs, and sheep remains limited due to ethical considerations and the difficulties associated with their maintenance and cost [13,14,15]. Small laboratory animals such as mice, rabbits, guinea pigs, and rats seem to meet these considerations and have already been used as animal models in postmenopausal osteoporosis research [16,17]. However, due to differences in rodent skeletal metabolism, their bone metabolism differs, which may limit translatability to human skeletal metabolism and constitutes a distinct challenge.

Given the number of animal models proposed, it is difficult to synthesize these studies to obtain a coherent understanding of the pathophysiology and select a reference animal model to test new therapies. Thus, herein, the aim of the present review is to provide a detailed overview of animal models of glucocorticoid-induced bone loss and explore how these models could be useful for preclinical and translational research on GIOP. We will also assess the quality of animal models by focusing on study design, drug dose, timing and size of the experimental groups, allocation concealment, and outcome measures.

## 2. Materials and Methods

### 2.1. Protocol and Registration

This systematic review was conducted and reported according to the PRISMA 2020 guidelines [18].

Our literature review was registered in the “Center for Reviews and Dissemination” PROSPERO; Registration number: CRD42021259669

### 2.2. Eligibility Criteria

The PICOS [18] (Population, Intervention, Comparator/Control Outcome, and Study design) concept was used to develop a search strategy.

Research has focused on animal studies of bone loss with glucocorticoid pharmacological treatment. In our review, we were interested in learning which animal models were the most relevant, robust, and reproducible. In addition, we focused on papers that measure, on one hand, the bone loss with either (dual-energy X-ray absorptiometry (DXA) or microtomography (µCT), and on the other hand, biomechanical properties to characterize the GIOP bone phenotype.

### 2.3. Information Sources

The PubMed database was searched between January and April 2021. The last search was performed on 15 April 2021.

### 2.4. Search Strategy

The studies were selected from a search strategy developed with an expert librarian on the PubMed database using the following keywords:

(osteoporo * OR “Osteoporosis” [Mesh] OR “bone loss” OR “glucocorticoid induced osteoporosis” OR “osteopenia” OR”Cancellous Bone” [Mesh] OR “Cortical Bone” [Mesh]) AND (“Glucocorticoids” [Mesh] OR “Methylprednisolone” [Mesh] OR “Dexamethasone” [Mesh]) AND (“models, animal” [Mesh] OR “Murinae” [Mesh] OR “Rabbits” [Mesh] OR “Sheep” [Mesh] NOT “Humans” [Mesh]) Filters: from 2011–2021

### 2.5. Selection Process

The inclusion criteria were as follows:

Experimental studies in animal models of glucocorticoid-induced osteoporosis or osteopenia in which the effects of glucocorticoid on biochemical markers, on bone tissues from the femur, the tibia or vertebrae were assessed through biochemical, biomechanical, histological, and imaging techniques. Studies were published in English and in internationally peer-reviewed journals between 2011 and 2021.

We excluded avian and zebra fish models as well as cellular models. Exclusion criteria also included studies related to osteonecrosis of the mandible or of the femoral head. We also excluded animal studies in which animals were also ovariectomized in order to limit our analysis and interpretation on GIOP mode. Ovariectomy in animals is a well-established model of bone loss in the literature for postmenopausal osteoporosis research [16,19,20].

### 2.6. Data Collection Process

Data were extracted into a template established before starting the searches and then verified by double reading.

GIOP was defined as any intervention study where animals were treated with GCs and resulted in alteration of calcified bone tissue.

Several variables for which data was collected were defined: animals that were treated with GCs compared to a placebo control and had at least one medical imaging (DXA, µCT) and either histomorphometry or biomechanical studies. Two investigators (AX and EL) independently assessed all the studies and consensus was reached through discussion with a third investigator (HT).

### 2.7. Data Items

Information was extracted from each of the articles according to PICOS on the following criteria: (1) characteristics of the animal models (species, strain, sex, age, weight, number of animals used); (2) type of intervention (glucocorticoids used, dose, frequency, duration, administration; versus placebo or versus another GCs; or versus another drug used for the treatment of osteoporosis); and (3) type of outcome measure (imaging techniques (DXA or µCT) and histomorphometry or biomechanical test (three point bending, micro-indentation, load test)).

### 2.8. Study Risk of Bias Assessment

This systematic review is exploratory in nature and aims to highlight the qualities and limitations of the species used for research in the GIOP. No risk of bias assessment was carried out. However, we assessed the methodological quality of the studies by referring to the work of Schulz et al. [21], particularly on the randomization of animal groups and statistical analysis.

### 2.9. Effect Measures

The studies should report DXA measurements expressed as BMC or BMD according to international recommendations. Micro-scanner or histomorphometry measurements should show microarchitecture parameters such as BV/TV, BS/BV, Tb.Th, Tb.Sp, Tb.N, as defined by Bouxsein and by Parfitt [22,23]. Finally, the strength tests must include at least one extrinsic parameter measurement [24,25].

### 2.10. Synthesis Methods

A study was eligible for our search if it included at least one animal model (described above) in which a glucocorticoid drug intervention (dexamethasone, methylprednisolone, prednisone, prednisolone, and cortisone) was performed compared to a placebo control group. The study also had to include imaging measurements of bone tissue (DXA or µCt or histomorphometry) or a strength test.

The selected studies were analyzed and classified in tables. If any of the information was missing or not clearly identified, the information was inferred from the article understanding or labeled not available (NA). For example, the number of subjects could be deduced from the results of the statistical tests.

## 3. Results

### 3.1. Study Selection

The search strategy identified 284 papers in the PubMed database. Based on title or abstract screening, 245 studies were excluded since they did not meet the eligibility criteria. The subsequent full-text assessment resulted in 39 records that were found eligible for the comprehensive review. Fourteen additional articles from our own personal records were included as they fell within the eligibility criteria of our work but did not emerge from the search strategy. Finally, 53 articles were considered (Figure 1).

### 3.2. Animal Characteristics

The majority of the studies selected and analyzed used rats (*n* = 34) as experimental models with a predominance for the Sprague Dawley strain. Studies using mice represented a third of the studies (*n* = 18) with a predominance of the C57BL/6 strain and its numerous knockout derivatives. One study used male C57BL/6 mice and male Sprague Dawley rats, which was counted twice in Table 1. Finally, one study in rabbit and one in sheep were included (Table 1).

Regarding the sex of the animals, in rats of all strains, there was no marked predominance. In mice, it appeared that males constituted the majority (55%). Finally, the few studies in rabbit and sheep did not allow us to establish a trend.

In Sprague Dawley rats (male and female), the age range was from eight weeks to six months with rats averaging three and a half months old. The age range varied from a few days (neonatal) to six months in the other strains.

In C57BL/6 mice, the age range was eight weeks to four months; in the other strains, the age range was seven weeks to four months. Age of rabbit (one study [26]) and sheep (one study [27]) is indicated in Table 2.

**Table 1 ijms-23-00377-t001:** Number of articles by species, strain, and sex in the GIOP model. One paper [28] included both mice and rats with different protocols.

	Rat (34 Protocols)			Mice (18 Protocols)			Rabbit (1 Protocol)	Sheep (1 Protocol)
	Sprague Dawley	Wistar	Albinos /LEW CrlCrlj	C57BL/6	C57BL/6 with Sprague Dawley	Other Strain	New Zealand White	Merino
Male	11	3	1	3	1	6	1	
Female	14	2	1	6		0		1
Male /Female	1	1				1		
Sex Not available						1		
Total articles	26	6	2	9	1	8	1	1

### 3.3. Induction of GIOP

The GCs used in murine models of GIOP are dexamethasone (DEX) (49%), prednisone (22%), and methylprednisolone (MP) (14%) in rats, and in mice, prednisolone (47%), MP (24%), and DEX (12%) (Figure 2).

The route of administration in rats was mainly subcutaneous (SC) (45%), intramuscular (IM) (22%), and Per. Os (P.O) (22%). The intraperitoneal (IP) and intravenous (IV) routes were less used (Table 3, Table 4 and Table 5. In mice, the most commonly used route was SC through pellets inserted surgically (pellet in subcutaneous: PSC) (61%). Other routes of administration were less used (P.O: 16%, SC: 16%) (Table 6 and Table 7). In rabbit and sheep, the routes of administration were IM and SC (Table 2).

In rats, DEX was administered at the lowest dose of 0.1 mg/kg daily for 60 days [29] and at the highest dose of 25 mg/kg twice per week for six weeks [30]. Prednisone was administered at the lowest dose of 1.5 mg/kg per day for 90 days [31] and at the highest dose of 6 mg/kg per day for 90 weeks [31]. In mice, prednisolone was administered at the lowest dose of 0.8 mg/kg per day for three weeks [32] and at the highest dose of 4 mg/kg per day for three weeks [32]. In rabbits and sheep, DEX was administered at a dose of 3 mg/kg twice weekly for 12 weeks [26] and prednisolone at a dose of 0.6 mg/kg five times weekly for seven months [27], respectively.

**Table 3 ijms-23-00377-t003:** Main characteristics of the proposed experimental protocol to induce GIOP in female Sprague Dawley. DEX = Dexamethasone, MP = Methylprednisolone, IM = Intramuscular injection, SC = Subcutaneous injection, IV = Intravenous injection, IP = Intraperitoneal injection, P.O = Per. Os, NA = Not Available.

References	Age	Weight	Molecule Used	Route of Administration	Dosage	Duration
2019 Y Xu [33]	8 W	250 ± 10 g	DEX	IM	2.5 mg/kg twice per week	2 M
2019 J Zhao [34]	3 M	280 ± 14 g	MP	SC	13 mg/kg 5 days per week	9 W
2018 Y Yang [35]	4 M	225 ± 25 g	Prednisone	P.O	5 mg/kg daily	14 W
2017 H Ren [36]	3 M	NA	DEX	SC	0.6 mg/kg every 3 days	3 M
2017 M Zhou [37]	6 M	200± 20 g	Prednisone	P.O	6 mg/kg daily	21 W
2017 G Chen [38]	4–5 M	250–275 g	Prednisone	P.O	5 mg/kg daily	90 D
2016 Z Chen [29]	NA	NA	DEX	SC	0.1 mg/kg daily	60 D
2016 Y Yang [39]	4 M	200–250 g	Prednisone	P.O	5 mg/kg daily	14 W
2016 G Shen [40]	3 M	NA	DEX	SC	0.6 mg/kg twice per week	3 M
2015 H Ren [41]	3 M	212 ± 30 g	DEX MP	SC	0.6 mg/kg twice per week 1 mg/kg daily	12 W
2013 M Khan [42]	NA	180 ± 20 g	DEX MP	IP SC	200 µg/kg 5 days per week 5 mg/kg 5 days per week	4 W
2017 G Pizzino [43]	5 M	250–275 g	MP	SC	30 mg/kg	60 D
2016 Y Jiang [44]	3 M	210 ± 20 g	DEX	IM	2.5 mg/kg twice per week	12 W
2016 D Liang [45]	4 M	NA	DEX	SC	0.6 mg/kg twice per week (prevention) 0.6 mg/kg daily (treatment)	3 M
2015 Y Liu [46]	12 W	263.5 ± 12 g	DEX	IV	2 mg/kg twice per week	12 W

**Table 4 ijms-23-00377-t004:** Main characteristics of the proposed experimental protocol to induce GIOP in male Sprague Dawley. DEX = Dexamethasone, MP = Methylprednisolone, IM = Intramuscular injection, SC = Subcutaneous injection, P.O = Per. Os, NA = Not Available, ^†^ Study using mice and Sprague Dawley rats.

References	Age	Weight	Molecule Used	Administration	Dosage	Duration
2021 Y Mo [28] ^†^	4 M	NA	Prednisone	P.O	5 mg/kg daily	16 W
2020 S Pal [47]	NA	260 ± 20 g	MP	SC	5 mg/kg daily	4 W
2019 L. Yang [48]	8 W	220 ± 10 g	DEX	IM	1 mg/kg twice per week	3 M
2014 M Feng [49]	6 M	220–240 g	DEX	SC	0.1 mg/kg daily	5 W
2013 Z Ren [50]	5 M	390 g	DEX	SC	0.1 mg/kg daily	5 W
2013 F-S Wang [51]	5 M	NA	MP	SC	5 mg/kg daily	1–2 or 4 W
2011 F-S Wang [52]	5 M	NA	DEX	SC	0.1 mg/kg daily	1–2 or 5 W
2012 L Cui [53]	6 M	390 ± 25 g	Prednisone	P.O	3.5 mg/kg daily	12 W
2017 Y Yang [54]	12 W	200 ± 20 g	DEX	IM	1 mg/kg twice per week	8 W
2014 S Lin [31]	3 M	300 g	Prednisone	P.O	1.5 mg/kg daily 3.0 mg/kg daily 6.0 mg/kg daily	90 D
2012 J-Y Ko [55]	4 M	NA	DEX	SC	0.1 mg/kg daily	1–2 or 5 W
2017 M Zhou [37]	6 M	220 ± 20 g	Prednisone	P.O	6 mg/kg daily	21 W

**Table 5 ijms-23-00377-t005:** Main characteristics of the proposed experimental protocol to induce GIOP in male and female rats (excluding Sprague Dawley). M = Male, F = Female, DEX = Dexamethasone, MP = Methylprednisolone, IM = Intramuscular injection, SC = Subcutaneous injection, PSC = Pellet in Subcutaneous, NA = Not Available.

References	Strain	Sex	Age	Weight	Molecule Used	Administration	Dosage	Duration
2020 Y Yang [56]	Wistar	F	6 W	180 ± 20 g	DEX	IM	2.5 mg/kg twice per week	7 W
2020 D Sato [57]	LEW CrlCrlj	F	5 W	125 g	Prednisolone	PSC	0.42 mg daily	6 W
2019 T Hou [58]	Albinos	M	Neo-natal	5–10 g	DEX	NA	0.1 mg/kg	6 W
2017 L.M.F. Lucinda [59]	Wistar	F	50 D	100–150 g	DEX	IM	7 mg/kg once per week	5 W
2016 N Han [30]	Wistar	M/F	3 M	283 ± 42 g	DEX	IM	25 mg/kg twice per week	6 W
2015 Z Achiou [60]	Wistar	M	19 W	450 g	MP	SC	5 mg/kg 5 days per week	9 W
2013 K Pichler [61]	Wistar	M	12 W	240 ± 20 g	Prednisolone	SC	7 mg/kg daily	4 W
2011 M Saito [62]	Wistar	M	6 M	330 g	Prednisolone	IM	10 mg/kg 5 days per week	4 W

**Table 6 ijms-23-00377-t006:** Main characteristics of the proposed experimental protocol to induce GIOP in male and female C57BL/6 and KO derivate. M = Male, F = Female, DEX = Dexamethasone, P.O = Per. Os, PSC = Pellet in Subcutaneous, SC = Subcutaneous injection, NA = Not Available, ^†^ Study using mice and rat Sprague Dawley.

References	Sex	Age	Weight	Molecule Used	Administration	Dosage	Duration
2021 Y Mo [28] ^†^	M	8 W	NA	Prednisone	P.O	2.1 mg/kg daily	8 W
2017 A Y Sato [63]	F	16 W	NA	Prednisolone	PSC	2.1 mg/kg daily	14 or 28 D
2016 A Y Sato [64]	F	4 M	25 ± 6 g	Prednisolone	PSC	1.4 mg/kg daily 2.1 mg/kg daily	90 D
2016 A Ersek [65]	F	12 W	NA	Prednisolone	PSC	2.5 mg	60 D
2019 Q Geng [66]	M	12 W	NA	DEX	SC	1 mg/kg 5 days per week	4 W
2019 L Mao [67]	F	10 W	20 ± 2.0 g	DEX	SC	10 mg/kg three times per week	90 D
2019 CG Fenton [68]	M	9 W	NA	Corticosterone	P.O	100µg/mL twice per week	4 W
2019 J D Schepper [69]	M	15 W	NA	Prednisolone	PSC	2.5 mg/kg daily	60 D
2018 C Ohlsson [70]	F	12 W	NA	MP	PSC	7.6 mg/kg daily	4 W
2018 I Bergström [71]	F	3 M	NA	Prednisolone	PSC	11 mg/kg daily	11 D

**Table 7 ijms-23-00377-t007:** Main characteristics of the proposed experimental protocol to induce GIOP in mice (excepted C57BL/6 and their KO derivatives). M = Male, F = Female, DEX = Dexamethasone, MP = Methylprednisolone, P.O = Per. Os, SC = Subcutaneous injection, PSC = Pellet in Subcutaneous, IP = Intraperitoneal injection, NA = Not Available.

References	Strain	Sex	Age	Weight	Molecule Used	Administration	Dosage	Duration
2021 A M Dubrovsky [72]	BALB/cJ	M	9 W	NA	MP	PSC	2.5 mg for 21 day/pellet	60 or 120 D
2019 S Adhikary [73]	BALB/C	M	8 W	22–25 g	MP	SC	10 mg/kg	10 W
2018 I Alam [74]	Col2.3- hWNT16TG	M/F	16 W	NA	Prednisolone	PSC	2.1 mg/kg daily	28 D
2017 G Mohan [75]	Swiss Webster	M	4 M	NA	Prednisolone	PSC	2.8 mg/kg daily	28–56 D
2016 F-S Wang [76]	129 S Npytm1RPA/J	M	16 W	NA	MP	IP	5 mg/kg daily	4 W
2015 W Yao [32]	dsRed-LC3 report	M	2 M	NA	Prednisolone	PSC	0.8–2.8–4 mg/kg daily	3 W
2011 M Marenzana [77]	BALB/c	NA	7 W	22.5 g	DEX	P.O	3 mg/kg daily	6 W or 9 W
2011 H Henneicke [78]	Col2.3-11βHSD2	M	8 W	NA	Corticosterone	PSC	1.5 mg/kg implanted at 0–1–14 or 21 days	28 D

### 3.4. Quality Assessment

Only 38 studies reported random assignment of animals to groups. Of these studies, 23% reported the method of randomization, either by animal weight or by initial BMD measurement. Only eight studies reported had blinded assessment of outcomes.

Although the majority of studies presented a quality protocol, it should be noted that 43% did not report the weight of the animals, 7% the age, 3% the number of subjects used in the protocol, and 2% did not specify the sex of the animals. A minority of studies did not clearly indicate the dose and duration of the treatment used. Protocols that used the P.O route did not always specify the mode of administration.

### 3.5. Techniques to Measure GIOP

The number of articles using DXA µCT histomorphometry and mechanical test by animal species and strain is reported in Table 8. In rats, 85% of the studies used mechanical testing, 82% DXA, 52% µCT, and 47% histomorphometry. In mice, 85% of the studies used µCT, 72% mechanical testing, 50% histomorphometry, and 44% DXA.

### 3.6. Bone Loss, Micro Architecture Alteration, and Decrease in Bone Strength Induced by GCs

We listed all significant results of bone loss microarchitecture alteration and decrease in bone strength induced by GCs assessed by DXA, µCT, histomorphometry, and biomechanical testing in Table 9. A total of 87% of the studies reported alterations in microarchitecture observed either by µCT or histomorphometry. Of these studies, 63% reported trabecular bone analysis, 28% trabecular and cortical bone analyses, and 8% cortical analysis.

## 4. Discussion

### 4.1. Characteristics of Species and Strains

The present systematic review provides an overview of animal models of GIOP. We found large heterogeneity in both methods applied to its establishment and in the animals used. Hence, the types of GC treatment, periods of administration, dose frequency, and administration route differ from study to study (Table 2, Table 3, Table 4, Table 5, Table 6 and Table 7). We limited our investigation to murinae, sheep, and rabbit. However, there is still heterogeneity in the models related to different species, strain, age, and genders used in the experimental protocol (Table 2, Table 3, Table 4, Table 5, Table 6 and Table 7).

Research into postmenopausal osteoporosis contributed to the use of rats as a reference animal model based on FDA guidelines [79], which have been widely published [16,17]. In the present work, 66% of the papers reviewed used rats, of which 77% involved Sprague Dawley (17% Wistar and 6% others). However, rats show some metabolic differences compared to human bone. The rat bone architecture does not have a Haversian system and some long bones retain their longitudinal growth capacity for most of their lives. Thus, beyond 30 months, the growth epiphyses of rats remain open [17]. In cancellous bones such as the lumbar vertebrae, the main activity before three months is bone shaping and not remodeling. These characteristics should be considered in dynamic histomorphometry analyses.

Although rats for many years have constituted a model of choice for bone studies, genetic engineering and biological tools remain limited for the study of this species. Conversely, mice who share more than 95% of their genome with the human species can be easily modified to create disease-specific knockouts. Mice in GIOP show a similar pattern of bone loss to humans, with an early phase of accelerated bone resorption followed by a slower phase of inhibited bone formation [80], but this is strain dependent [65]. In fact, it has been shown that mice with a C57Bl/6 genetic background have a lower susceptibility to GIOP than CD1 strains [65].

Because of their size, sheep are the preferred model for research into the treatment of osteoporotic fracture and for studies on biomaterials for medical devices. The most commonly used osteoporotic sheep model is the ovariectomized sheep, but this model alone appears to have moderate impacts on bone mass as the majority of studies combine the use of OVX with GC treatment [81]. However, models of osteoporosis induced by GCs alone appear to approximate bone conditions comparable to those found in steroid-treated humans with changes in microarchitecture and mechanical properties [13,82]. Only one study [27] investigated GIOP in a purely GCs induced osteoporosis protocol (without ovariectomy) that showed an increase in the cortical porosity. However, this study combined the effects of GCs with a diet low in calcium and phosphorus, which is a source of bias in the study of bone loss due to steroid therapy alone.

Adult rabbits have a Haversian system and reach skeletal maturity. In addition, they have a high turnover rate with remodeling predominating over the shaping process, all conditions that can be considered as promising to constitute a model for osteoporosis research. In addition, rabbits are sexually mature at 6–8 months of age and exhibit closure of the growth epiphyses [83]. However, achieving significant bone loss in rabbits necessitates concomitant ovariectomy [84].

Rats and mice are the most commonly used species in GIOP animal studies. These animals are easy to breed, house, and manipulate and have a relatively low operating cost, unlike large animals such as sheep.

### 4.2. Molecules, Dose, Duration, and Route of Administration

Several GCs have been used in animal models of GIOP (Figure 2). Although, DEX is the most widely used corticosteroid in rats and prednisolone is the most widely used corticosteroid in mice, it is difficult to consider them as the reference GCs to be used in animal models of GIOP as other molecules such as prednisone and MP appear to be effective in inducing GIOP (Table 10). However, these corticosteroids do not have the same glucocorticoid potency as their natural counterpart, hydrocortisone. Their mineralocorticoid activity is negligible. DEX has 25 times the activity of hydrocortisone, MP five times, and prednisolone four times [85]. The potency of GCs has been studied in a single protocol that compared the effects of DEX and MP (at a dose of 0.6 mg/kg twice weekly and 1 mg/kg once daily, respectively) but for the same duration (12 weeks), the same route of administration, and the same strain [41]. In this study, the loss of BMD measured on the L1–L3 vertebrae was greater in the DEX group compared to the MP group [41]. Interestingly, in another study aiming at comparing DEX (200 µg/kg 5 day a week IP) vs. MP (5 mg/kg 5 day a week SC) in Sprague Dawley female rats with the same duration, the comparison could not be performed due to a high rate of death in the DEX group. However, the dose used in this protocol was lower than the usual doses reported in other papers using Sprague Dawley females (Table 2).

It should be noted that the relative efficacy of GCs may vary depending on the route of administration and dosage, and that the effects may differ depending on the site of measurement. One study investigated the dose effect with prednisone on bone metabolism using different doses (1.5, 3, or 6 mg/kg/day for 90 days P.O.) in male Sprague Dawley rats [31]. The results showed that for a dose lower than 6 mg/kg, the measurement of Tb.Th at the proximal tibia was not significantly different from the control group. However, for the same dose (6 mg/kg), the result of Tb.Th was not significantly different from the control at the femur, illustrating the fact that for the same dose, the sensitivity to GCs could be different according to the bone site of interest.

In clinical routine, the main routes of administration are P.O, IM, and IV [86]. To mimic GIOP, it would therefore be logical to follow these routes of administration. However, our studies have highlighted other routes of administration such as the IP and PSC.

Daily injections are stressful for the animals, which may interfere with the study and bias the results. Repeated injections have been shown to increase serum corticosterone levels one hour after injection in mice [87]. We observed in mice that those of PSC were predominantly employed (66% of studies in mice). However, the use of PSC involves surgery, which requires increased attention to animals and more hover increases the risk of postoperative infection. The majority of the studies in this work reported that GCs were given by injection and not P.O, which is opposite to human clinical practice [86]. However, the way in which the P.O was given was not always explained. Administration can be by capsule in the feed, dissolved in the drinking water or by gavage to ensure absorption of the GCs, but the latter method induces additional stress for the animal [88].

There was also high heterogeneity in the duration of GC treatment in the studies analyzed in the present review. In Sprague Dawley male DXA, analyses showed that GC treated rats (MP 5 mg/kg daily SC) had lower BMD and BMC than the vehicle-treated group as soon as after two weeks of treatment. However, it needs four weeks of GC treatment to observe a reduced metaphyseal trabecular microstructure [51]. In two other studies, the same strain and sex (male Sprague Dawley) rats were treated with 0.1 mg/kg/day SC DEX for one, two, or five weeks. The µCT measurements showed that the rats had more metaphyseal trabecular bone loss than the vehicle-treated group as soon as five weeks of treatment [52,55]. DXA analyses showed a reduction in BMD at both two [55] and five weeks [52,55]. These findings illustrated the different sensitivity of the two devices (DXA, µCT) to show the GC effects and could give some information about the minimal length of treatment to induce GIOP. However, the aforementioned studies seem to show that DXA measurement should evidence a bone decrease earlier than with µCT measurement in Sprague Dawley rats, which cannot be extrapolated to other animals. Hence, in female six week old C57BL/6 mice, TB.Th measured by µCT was significantly decreased at both distal femur and proximal tibia after 14 days of prednisolone 2.1 mg/kg/d, but a significant decrease in BMD (measured by DXA) at the femur and spine was only observed after 28 days of treatment [63].

### 4.3. Impact of the Weight, Age, and Sex on the Establishment of GIOP Model

GCs in animal models affect both body weight and size. Thus, theoretically when measuring GC effects on bone loss, bone size adjustments should be performed to limit bias in interpreting the effect on BMD [89]. It appears that none of the studies found in this review reported an adjustment for bone size in the section on statistical analysis.

Conversely, findings from a study using prednisone in young rats (three months) reported data of Lin et al. in 2014 [31], indicating that prednisone reduced bone growth, which raises the question of whether it is appropriate to use rats that are barely skeletally mature. Rats reach sexual maturity at about six weeks of age, but their bodies including their skeletons are constantly growing. Rats are considered adult when they are socially mature, which is six months later [90]. It is accepted that one month of rat life corresponds to 2.5 years of human life [91].

Gender and strain-specific efficacy of GCs have been previously demonstrated [92,93]. The effects of prednisone (6 mg/kg/day P.O) in male and female Sprague Dawley rats were investigated on bone parameters for 21 weeks [37]. Findings demonstrated that the BV/TV parameter decreased by 52% and 27.0% at the femur and L4 vertebra, respectively, in females, whereas in males, the loss was 28.6% and 14.0% at the femur and L4, respectively. Unfortunately, in a study conducted in male and female Wistar rats to demonstrate the efficacy of a Chinese herbal medicine in GIOP, no comparison was undertaken to compare males and females concerning the results on bone parameters [30].

### 4.4. Methods of Measurement of the Establishment of the GIOP Bone Phenotype

Laboratory techniques are used in GIOP animal models and inside these techniques, their nuances are one of the key issues related to the appropriate methodology to evaluate animal models of GIOP and extrapolate findings from animals to humans.

In the present systematic review, we deliberately decided to report original articles using either dual-energy X-ray absorptiometry (DXA) or micro-computed tomography (µCT) to assess areal BMD or volumetric BMD to demonstrate GIOP bone loss. We also chose to consider articles that evaluated the effects of GCs on bone microarchitecture either by µCT or by quantitative histological techniques. In addition, to appreciate the most appropriate GIOP models, we sought to select studies that evaluated the bone biomechanical properties. Indeed, in order to actually mimic GIOP in humans, we considered that a relevant animal model of GIOP would produce both bone loss and alterations in the microarchitecture, but also decrease the bone biomechanical properties.

DXA is considered the gold standard for the diagnosis of human osteoporosis. This technique is also widely used in rodents and provide assessments of bone mass and gross morphology in and ex vivo. Although measurements were performed on small bones or on the whole body, the DXA measurement was accurate and precise. However, this technique is limited by its low spatial resolution; moreover, the accuracy of this technique cannot permit compartment specific bone parameter analyses. However, DXA and peripheral quantitative CT produced comparable results in terms of precision, accuracy, and sensitivity to change when examining the rat femur [94].

Conversely, µCT is currently considered as the gold standard for the assessment of vBMD and microarchitecture, both at the trabecular and cortical bone [22]. Furthermore, new devices provide the opportunity of in vivo scanning that permit longitudinal analyses such as the monitoring of bone metastasis [95] or bio-integration of bone implants [96]. In the present systematic review, 64% of the studies were conducted with ex vivo µCT and 1% with the functionality of in vivo scanning.

Histological analyses are very important because they provide both static and dynamic histomorphometric parameters and information at the tissue and cellular levels. Using this method, both cancellous bone (at lumbar vertebral bodies, tibia, or femur metaphysis) and the cortical bone can be investigated. The cortical bone is mainly assessed at the diaphysis of the long bone.

However, even in the best GIOP animal models, spontaneous fracture such as in humans does not occur. Consequently, biomechanical testing has been considered in GIOP animal models as a reliable surrogate marker of bone fragility [97,98].

Bone strength is usually evaluated through the three-point bending test of long bone (femur or tibia) and axial compression tests on vertebral bodies. However, this review provides evidences that a number of biomechanical properties were reported, hence, more than forty different parameters were provided, as indicated in Table 9. The heterogeneities in the methods justify the demand of consensus regarding both extrinsic and intrinsic biomechanical parameter methods are reported in the literature. Mechanical tests in three-point bending and compression determine the actual strength of the bone material. Depending on the load applied, the stresses can be compressive, tensile, or shear forces. The parameters measured reveal the extrinsic (structural) forces of the bone: ultimate load (N), stiffness (N/mm), yield load (N), elastic energy (N.mm), and plastic energy (N.mm). The intrinsic force can be described by the following parameters: Young’s modulus (modulus of elasticity MPa), toughness (MJ/m^3^), and ultimate stress (N/mm^2^) as defined by Turner and Draca [24,25]. Sometimes, solely the units of bone biomechanical properties have been presented without a full description of the parameter evaluated.

It is well known that GIOP patients fracture at higher BMD values than in postmenopausal osteoporosis. In clinical practice, fracture is most often associated with altered density and micro-architecture; however, in GIOP, the epidemiological data seem to show that this relationship is not always true [99]. Interestingly, only one study in our review reported a decrease in bone strength (maximum load (N) and energy (mJ)) measured at the femur while there was no significant decrease in BMD measured at the femur [62].

Finally, several studies are in accordance with the present good model definition of GIOP (i.e., a loss of bone density) with an alteration in the micro-architecture that leads to a decrease in the biomechanical strength of the bone (Table 10).

### 4.5. Methodological Quality

We also analyzed the methodological quality of all eligible reports using the definition by Schulz et al. [21] and adapted by Perel et al. [100]. In those animal research studies [21,100], the emphasis was placed on random allocation to group, adequate allocation concealment, and blinded assessment of outcome.

Allocation of random animals to experimental groups was performed in two thirds of the eligible studies and only eight studies [42,43,57,61,63,64,69,71] performed a blinded assessment of outcomes. Furthermore, this blinded assessment was only conducted for histomorphometric measurements. Methods for adequate allocation concealment were not clearly reported in the studies analyzed in our work. This lack of adequate allocation concealment has already been evidenced in both antenatal corticosteroids and corticosteroids for traumatic head injury studies where this was reported in 0% and 18% of the studies, respectively [100].

In addition, age, gender, weight, techniques of P.O administration of GCs, and number of animals were not systematically reported in some studies.

Finally, we did not find any study reporting the a priori sample size calculation of animals in the statistical methods section.

However, we acknowledge that authors might have undergone good quality procedure in conducting their study without correctly reporting it in their article.

### 4.6. Strength and Limitation

The strength of the current review is warranted by our systematic approach to search for original articles in one of the major online databases (i.e., PubMed).

All papers identified by the search strategy were retrievable and analyzed. However, we acknowledge that there are some limitations to consider. First, we limited our search strategy to one database, thus articles not indexed in PubMed were not included and reviewed even though they could potentially be relevant for our purpose. Second, we limited our search strategy to the last decade and focused our work on a limited number of species, omitting models using dogs or emerging zebra fish models [101,102].

Nevertheless, 53 full text articles were analyzed.

It is widely accepted that the underlying inflammatory disease that require GCs has a role in the pathophysiology of GIOP in human [103]. However, in our protocol, we decided to exclude studies that designed their methodological protocols including a previous induction of a glucocorticoid requiring disease.

Attempts should be made to standardize pre-clinical GIOP animal models. Such guidelines would enhance cross-laboratory comparisons and avoid the selection of the animal model, which could be driven more by convenience or past experience rather than based on evidence.

We have to underline that in the ideal, GIOP animal models would exhibit disease characteristics that are comparable to the human conditions. However, first, none of the eligible studies designed their methodological protocols including a previous induction of a glucocorticoid requiring disease. Indeed, it is widely accepted that the underlying inflammatory disease that requires GCs has a role in the pathophysiology of GIOP in humans [103]. This is a limitation of our study since we did not want to analyze the combined effects of corticosteroids on bone with pathology or treatments that could induce a bone phenotype.

Furthermore, we only found one study [45] that proposed, as in the human GIOP clinical trials [104], a designation of both treatment and prevention.

## 5. Conclusions

Direct comparison among studies is challenging due to the heterogeneity of the various experimental designs reported in relation to the dose, the route of administration, the duration, the type of GCs, and possibly the different bioavailabilities of the different GCs used as a function of the animal species. Nevertheless, we showed that the use of DEX in Sprague Dawley rats and prednisolone in mice are the most popular GIOP models, and that the dose and duration of these last GCs should be decided according to the age of the animal and the bone site chosen to be assessed. The results of this systematic review suggested that it lacks studies on the length of the persistence of the deleterious effects of GCs on bone tissue. Such studies might improve the management in terms of duration of therapies against GIOP. Although the present systematic review provides relevant information on the animal literature in GIOP, it also points to the poor quality of some protocols and suggest the need for guidelines in GIOP animal models to improve standardization and research outcomes aiming at the establishment of a better predictability of animal research in further human clinical trials. 

## Figures and Tables

**Figure 1 ijms-23-00377-f001:**
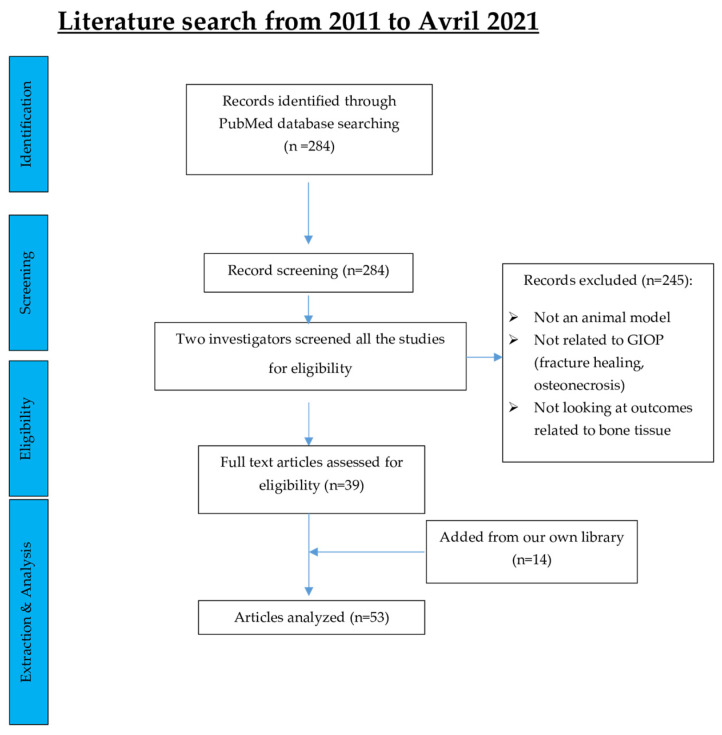
Flowchart illustrating the study selection process.

**Figure 2 ijms-23-00377-f002:**
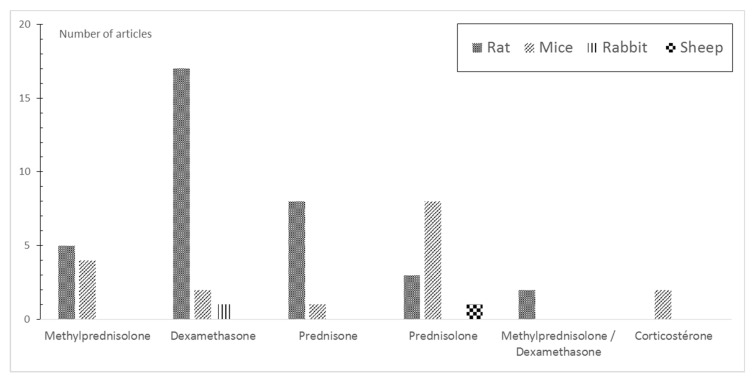
Distribution of glucocorticoid use by species and number of articles.

**Table 2 ijms-23-00377-t002:** Main characteristics of the proposed experimental protocol to induce GIOP in rabbit and sheep. M = Male, F = Female DEX = Dexamethasone, IM = Intramuscular injection, SC = Subcutaneous injection.

References	Species	Strain	Sex	Age	Weight	Molecule Used	Administration	Dosage	Duration
2014 Z Yongtao [26]	Rabbit	New Zealand White	M	32 w	3.2 ± 0.45 kg	DEX	IM	3 mg/kg twice per week	12 W
2011 M Ding [27]	Sheep	Merino	F	4–6 Y	55 ± 10 kg	Prednisolone	SC	0.60 mg/kg 5 times per weeks	7 M

**Table 8 ijms-23-00377-t008:** Number of studies by species and strain using bone measurement techniques.

	Rat		Mice		Rabbit	Sheep
	Sprague Dawley	Other Strain	C57BL/6	Other Strain	New Zealand White	Merino
DXA	20	8	5	3	1	
µCT	17	1	8	7		1
Mechanical Test	23	6	6	7	1	1
Histomorphometry	13	3	6	3	1	

**Table 9 ijms-23-00377-t009:** Results of bone quality assessments reported by the studies. The DXA parameters are BMD (bone mineral density) and BMC (bone mineral content). The µCT parameters are BV/TV (bone volume/tissue volume), BS/BV (bone surface/bone volume), Tb.N (trabecular number), and Tb.Th (trabecular thickness), Tb.Sp (trabecular separation), SMI (structure-model index). The used static histomorphometry parameters are: Oc.N/BS (osteoclast number/bone surface), Ob.N/BS (osteoblast number/bone surface), Oc/S/BS (osteoclast surface/bone surface), Ob.S/BS (osteoblast surface/bone surface), ES/BS (eroded surface/bone surface), Oc.Pm/B.Pm (osteoclast perimeter/bone perimeter), Ob.Pm/B.Pm (osteoblast perimeter/bone perimeter), OS/BS (osteoid surface/bone surface). The dynamic histomorphometry parameters are: MS/BS (mineralizing surface/bone surface), MAR (mineral apposition rate), BFR/BS (bone formation rate/bone surface).

References	Bone Loss Measuring by DXA	Alterations of the Microarchitecture Observed by µCT or Static/Dynamic Histomorphometry	Loss in Bone Strength Parameters Observed by Biomechanical Testing
2019 Y Xu [33]	Yes at femur (BMC BMD)	Yes at femur (BV/TV, Tb.Th, Tb.Sp, Tb.N) by µCT	Yes at femur (maximum stress (MPa), maximum load (N), elasticity modulus (N/mm^2^)
2019 J Zhao [34]	Yes at femur and L5 vertebra (BMD)	Yes at proximal tibia (BV/TV, Tb.Th, Tb.Sp, Tb.N, %L.Pm, MAR, BFR/BV, BFR/TV, BFR/BS, Oc.N/BS, Oc.Pm/BS) by histomorphometry	NA
2018 Y Yang [35]	NA	Yes at proximal femur (Tb.Ar, Tb.Th, BV/TV, Tb.Sp, SMI, DA) by µCT	Yes at femur (elastic load (N), bending energy (N x mm)
2017 H Ren [36]	Yes at L1–L5 vertebrae (BMD, BMC, AREA)	Yes at L2 vertebra (BS/TV, BV/TV, Tb.Th, Tb.Sp, Tb.N, vBMD) by µCT	Yes at L2 vertebra (compressive strength (N), compressive displacement (mm), energy absorption capacity (J))
2017 M Zhou [37]	Yes (BMD BMC at the L4 vertebra, whole femur in male; BMD proximal femur, BMC distal femur in female)	Yes (at the distal femur in female: BV/TV, density, SMI, Tb.N, Tb.Th, Tb.Sp; at the distal femur in male: BV/TV, density) by µCT	Yes (at the femur in male elastic load (N), stiffness (N/mm) at the femur in female: elastic load (N); at the L5 vertebra in female: elastic load (N), maximum load (N), break load (N), stiffness (N/mm))
2017 G Chen [38]	Yes at femur (BMD, BMC)	Yes at proximal tibia (Tb.Th, SMI by µCtT, Tb.Ar, Tb.N, Tb.Th, Tb.Sp by histomorphometry)	Yes at femur (maximum load (N), Breaking load (N), Yield load (N), bending energy (mJ))
2016 Z Chen [29]	Yes at proximal femur (BMD)	NA	Yes at femur (ultimate load (N), Stiffness (N/mm))
2016 Y Yang [39]	NA	Yes at proximal femur (BV/TV, Tb.Th, Tb.Sp); at proximal metaphysis tibia: %L.Pm, BFR/TV) by histomorphometry	Yes at femur (fracture load (N), Bending energy N x mm))
2016 G Shen [40]	Yes at L1–L3 vertebrae (BMC, BMD, AREA)	Yes at L4 vertebra (BS/TV, BV/TV, Tb.Th, Tb.Sp, Tb.N, vBMD, SMI) by µCT	Yes at L4 vertebra (compressive strength (N), compressive stiffness (N/mm),compressive displacement (mm), energy absorption capacity (N))
2015 H Ren [41]	Yes at L1–L3 vertebrae (BMC, BMD, AREA)	Yes at L4 vertebra for DEX group and MP group (BV/TV, BS/TV, SMI, Tb.Th, vBMD) only DEX group (Conn.D, Tb.Sp, Tb.N) by µCT	Yes at L4 vertebra for DEX group and MP group (compressive strength(N))
2013 MP Khan [42]	NA	Yes at hypophysis/diaphysis femur and tibia (vBMD, BV/TV, Conn.D, SMI, Tb.Th, Tb.Sp, Tb.N, Porosity, DA) by µCT; (Ct/th, MAR, pBFR/BS) and by histomorphometry	Yes at femur (ultimate load (N), Energy (mJ), Stiffness (N/mm))
2017 G Pizzino [43]	Yes at femur (BMD)	Yes at femur and vertebra (BV/TV, Tb.Th) by µCT	Yes at femur (maximum load (N))
2016 Y Jiang [44]	No statistical difference reported at femur and whole body (BMD, BMC)	Yes at L4 vertebra (Tb.Ar, Tb.N, Tb.Sp) by histomorphometry	NA
2016 D Liang [45]	Yes at L1–L4 vertebrae (BMC, BMD)	Yes at L2 vertebra (BV/TV, SMI, Tb.N, Tb.Th, Tb.Sp, vBMD) by µCT	Yes at L2 vertebra (compressive strength (N), compressive displacement (mm), energy absorption capacity (J))
2015 Y Liu [46]	Yes at femur (BMD) and no statistical difference reported at L5 vertebra	Yes at proximal tibia by µCT (BV/TV, Tb.Th, Tb.N, Tb.Sp) and by histomorphometry (BV/TV, Tb.N, Tb.Sp, MS/BS, MAR, BFR/BS, BFR/BV, OcS, OcS/BS)	Yes at femur (energy (J), Bending stiffness (N/mm))
2021 Y Mo [28]	NA	Yes in rat at distal femur (vBMD) by µCT; at proximal tibia: %Tb.Ar, Tb.Wi, Oc.N, %Oc.Pm, %Ob.Pm) by histomorphometry; and in mice at distal femur (VBMD, Tb.Th) and by µCT	Yes at femur in rat (maximum load (N), fracture load (N), stiffness (N)) and in mice (elastic load (N))
2020 S Pal [47]	NA	Yes at femur (vBMD, BV/TV, Tb.N, Tb.SP, Tb.Th, SMI, BMD, Ct.Th, Periosteal perimeter) at L5 vertebra (vBMD, BV/TV, Tb.N, SMI) by µCT	Yes at femur (peak load (N), energy (mJ), stiffness (N/mm))
2019 L. Yang [48]	NA	Yes at vertebra (BMD, TMD, Conn.D, Tb.Th, Tb.Sp) at femur (BMD, TMD, Conn.D, Tb.Th, Tb.Sp, Tb.N, BV/TV) by µCT	Yes at femur (Bending load (N), Elastic modulus (MPa))
2014 M Feng [49]	Yes at femur (BMD)	Yes at proximal femur (BFR/BF, N.Ot, N.Ob, BV/TV, Tb.Th, N.OC/BS, Tb.Sp) by histomorphometry	NA
2013 Z Ren [50]	Yes at femur (BMD)	Yes at proximal femur (BV/TV, Tb.N, Tb.Th, Tb.Sp) by µCT	Yes at femur (Peak load (N))
2013 F-S Wang [51]	Yes at femur (BMD, BMC)	Yes at femur (BV/TV, Ct. Porosity) by µCT	Yes at femur (Load (N))
2011 F-S Wang [52]	Yes but bone site no reported (BMD)	Yes at proximal tibia (BMC) by µCT; (BFR/BS, BV/TV, Ob surface, Oc surface) and by histomorphometry	Yes but bone site no reported (Peak load (N))
2012 L Cui [53]	Yes at proximal and whole femur (BMD), measured by single photon	Yes at proximal tibia (BV/TV, Tb.Wi, ObS/BS, LGR, MAR, BFR/TV, Ec.MS/BS, EC.MAR, Ec.BFR/BS) by histomorphometry	Yes at femur (maximum force (N), maximum deflection (mm))
2017 Y Yang [54]	Yes at whole femur (BMD)	Yes at proximal metaphysis femur (Tb.Ar, Tb.Th, Tb.Sp, Tb.N) by histomorphometry	Yes at femur (maximal load (kg), ultimate deflection (mm))
2014 S Lin [31]	Yes at whole femur (BMD (Significant result at 6 mg/kg/d prednisolone in femur))	Yes at proximal tibia (Tb.Th, MS/BS, MAR, BFR/BS, BFR/BV) (significant result at 6 mg/kg/d prednisolone in tibia); Ob.S/BS, Oc.S/BS (significant result for doses below 6 mg/kg/d prednisolone in tibia); (Tb.Th, Tb.N, Tb.Sp (significant result at 3 mg/kg/d prednisolone in femur); (Ps MAR, Ps.BFR/BS (significant result at 3 mg/kg/d prednisolone in tibia shaft)(Ct.Th, Ec.MS/BS, Ec.MAR, Ec.BFR/BS (significant result at all dose)) by histomorphometry	Yes at femur and L5 vertebra (elastic load (N), maximum load(N) fracture load (N) stiffness coefficient (N/mm) (Significant result at 6 mg/kg/d prednisolone in femur)); maximum load (N), Young’s modulus at L5 vertebra (MPa))
2012 J-Y Ko [55]	Yes but bone site no reported (BMD, BMC)	Yes (BMC) by µCT; (BV/TV, Ob.S/BS, Oc.S/BS, BFR/BS) and by histomorphometry	Yes at tibia (peak load (N)
2020 Y Yang [56]	Yes at femur (BMD, BMC (measured by single photon bone mineral density analyzer))	Yes at femur (trabecular area index) by histomorphometry	NA
2020 D Sato [57]	Yes at distal femur (BMD)	Yes at proximal tibia (Oc.Pm/B.Pm, ES/BS, N.Oc/BS, OS/BS, Ob.S/BS, BFR/BS) by histomorphometry	Yes at tibia (maximum stress (N/cm^3^))
2019 T Hou [58]	Yes at proximal tibia (BMD)	NA	Yes at femur (peak load (N), ultimate stiffness (N/mm))
2017 L.M.F. Lucinda [59]	Yes (BMD) at tibia	NA	Yes at tibia (maximum load (N), bone stiffness (N/m), energy (mJ)
2016 N Han [30]	Yes (BMC, BMD) at femur	NA	Yes at femur (Flexure strength (Mpa), maximum bending force (N))
2015 Z Achiou [60]	Yes (BMC, BMD) at femur	Yes (BV/TV, Tb.N, Tb.Th, Tb.Pf, Ct.Ar, Ct.Th) at femoral mid-diaphysis by µCT	No statistical difference measured
2013 K Pichler [61]	Yes (BMD) at whole body, vertebra, and femur	No statistical difference reported at femur by histomorphometry	NA
2011 M Saito [62]	No statistical difference reported	NA	Yes at femur (maximum load (N), energy (mJ))
2017 A Y Sato [63]	Yes (BMD (at whole body at 14 days and whole body, femur, L1–L6 vertebrae at 28 days)	Yes (Tb.Th) at distal femur and proximal tibia by µCT; (BFR/BS, MAR, MS/BS) proximal tibia (periosteal and endocortical) and by histomorphometry	NA
2016 A Y Sato [64]	Yes (BMD) at whole body and L1–L6 vertebrae	Yes (Tb.Th, BA/TA, total Ct.Th, dorsal Ct.Th) by µCT at L6 vertebra; (MAR, BFR/BS, N.Oc/BS, Oc.S/BS) and by histomorphometry at L1–L3 vertebrae	Yes at L6 vertebra (ultimate force (N), energy to ultimate load (mJ), toughness (mJ/mm^3^))
2016 A Ersek [65]	NA	Yes (BV/TV, Tb.N, Tb.Pf, SMI, Ct.Th) at vertebra by µCT (N.Oc/T.Ar, Oc.S/BS) at vertebra and by histomorphometry	Yes at femur (maximum load (N), elastic modulus (MPa))
2019 Q Geng [66]	Yes (BMC) at total body, vertebrae, and femur	Yes (BV/TV, Tb.N, Tb.Th, Ct.Th, Ct.V, SMI, Tb.Sp) at femur by µCT; (BV/TV, T.Col, N.Ob/BS, Ob.S/BS, MS/BS, MAR BFR/BS) at distal femur and by histomorphometry	Yes at femur (maximum load (N), energy absorption (N x mm), stiffness (N/M), ultimate displacement (µm), yield displacement (µm), yield load (N))
2019 L Mao [67]	Yes (BMD) at femur	NA	Yes at tibia (ultimate load, stiffness)
2019 CG Fenton [68]	NA	Yes (BV/TV, Tb.N, Tb.Th, Tb.Sp) at tibia by µCT; (N.Ob/B.Pm) at L3–4 vertebrae by histomorphometry	NA
2019 J D Schepper [69]	NA	Yes (BV/TV/BW, Tb.Sp, Tb.Th, Tb.N, BV/TV) at femur by µCT	No statistical difference reported
2018 C Ohlsson [70]	Yes (BMC) at total body	Yes (BV/TV) at femur by µCT; (Tb.Th, MAR, Ct.Th, Endosteal circumference, Ct.Po, BFR) at femur and by histomorphometry	NA
2018 I Bergström [71]	NA	Yes (cBMC, Ps.Pm, Imoment of inertia, moment of resistance, Ct.Ar, Ct.Th) at proximal tibia by µCT	No statistical difference reported
2021 A M Dubrovsky [72]	NA	Yes (Ct.Ar, Ct.Ar/Tt.Ar, CT.Th) at central femur; (Tb.Th) at distal femur by µCT	Yes (Yield load (N)) at femur; (ultimate load (N), Yield load (N), work to ultimate force (N mm) for 120 days treatment at L6 vertebra.
2019 S Adhikary [73]	NA	Yes (vBMD, BV/TV, Tb.N, Conn.Den, Tb.Sp, SMI) at epiphysis femur; (CT.Ar, T.Ar, T.Pm) at diaphysis femur; (vBMD, Ct.Th, T.Ar, B.Ar, T.Pm, MMI) at disphysis tibia by µCT	Yes at femur (stiffness (N), energy (mJ), power (N))
2018 I Alam [74]	Yes (aBMD, BMC (only in female)) at femur	Yes (only in Female) (Tb.N, Tb.Sp) at L5 vertebra; BA/TA, Ct.Th, pMOI,) at femur by µCT	Yes at femur (stiffness, ultimate force, energy to ultimate force (only in female))
2017 G Mohan [75]	NA	Yes (BV/TV, Ct.BV (only treatment study)) at distal and mid-shaft femur by µCT	Yes (maximum load (prevention (28 days) and treatment (56 days) study)) at L6 vertebra; (maximum load (only treatment study) at femur
2016 F-S Wang [76]	Yes (BMC, BMC) at femur	Yes (B.Ar/T.Ar, Tb.Th, Tb.N, Tb.Sp, Bv/TV, BFR/BS, Ob.S, Oc.S) at femur by histomorphometry	NA
2015 W Yao [32]	NA	Yes (BV/TV) by µCT; (Tb.Th, Conn.D (at 4 mg/kg/d prednisolone)) at L5 vertebra by histomorphometry; (BMD, Ec-MS/BS, Ps-BFR at 2.8 mg/kg/d prednisolone, BV, Ec-MS/BS, Ec-BFR, Ps-MS/BS, Ps-BFR at 4 mg/kg/d prednisolone) at mid femur and by histomorphometry	Yes (maximum load (N), apparent ultimate stress (Mpa), at 4 mg/kg/d prednisolone apparent toughness (kj/m^2^) at 4 mg/kg/d prednisolone)at vertebral; (apparent ultimate stress (Mpa), apparent toughness (kj/m^2^)) at 4 mg/kg/d prednisolone at femur
2011 M Marenzana [77]	Yes (BMD) at femur	Yes (Tb.Th, Tb.N) at distal femur TB.Th, longitudinal length) at L5 vertebra (Ct volume, metaphysis Tb.Th) at femur by µCT	Yes at femur (maximum load (N), ultimate strength)
2011 H Henneicke [78]	NA	Yes (Tb.Th, Ct.Th, CT.Ar, MAR, BFR, pericortical Area, osteoclast/pericortical surface, pericortical area) at tibia by µCT and by histomorphometry	No statistical difference reported in mechanical load (N) and elastic modulus (Mpa)
2014 Z Yongtao [26]	Yes (BMD at 12 weeks treatment) at L3–L4 vertebrae	Yes (BV/TV, Tb.Th, MS/BS, MAR, BFR/BS, N.Oc/BS, Oc/BS, ES/BS) at L3 vertebra by histomorphometry	Yes at L4 vertebra (maximum load (N), stiffness (N/mm), fracture stress (N/mm^2^))
2011 M Ding [27]	NA	Yes (CT.Po, bone surface: volume ratio, bone surface density, cross sectional area followed by 3 months without treatment) at midshaft femur by µCT	No statistical difference measured in ultimate stress (MPa), ultimate strain (%) Young’s modulus (GPa), failure energy (kJ/cm^3^) at femur

**Table 10 ijms-23-00377-t010:** Studies responding to our definition of a good GIOP animal model. SD = Sprague Dawley.

References	GCs Used	BMD Loss	Alteration of the Microarchitecture	Decrease in Biomechanical Properties
2019 Y Xu [33] Rat female SD	DEX (2.5 mg/kg twice per week for 2 months, IM)	Femur	Femur	Femur
2017 H Ren [36] Rat female SD	DEX (0.6 mg/kg every 3 days for 3 months, SC)	L1–L5 vertebrae	L2 vertebra	L2 vertebra
2017 M Zhou [37] Rat female SD	Prednisone (6 mg/kg daily for 21 weeks, SC)	L4 vertebra and femur	Femur	Femur
2017 G Chen [38] Rat female SD	Prednisone (5 mg/kg daily for 90 days, P.O)	Femur	Tibia	Femur
2016 G Shen [40] Rat female SD	DEX (0.6 mg/kg twice per week for 3 months, SC)	L1–L3 vertebrae	L4 vertebra	L4 vertebra
2015 H Ren [41] Rat female SD	DEX (0.6 mg/kg daily for 12 weeks, SC) MP (1 mg/kg daily for 12 weeks, SC)	L1–L3 vertebrae	L4 vertebra	L4 vertebra
2017 G Pizzino [43] Rat female SD	MP (30 mg/kg for 60 days, SC)	Femur	Femur and vertebra	Femur
2016 D Liang [45] Rat female SD	DEX (0.6 mg/kg twice per week or daily for 3 months, IM)	L1–L4 vertebrae	L2 vertebra	L2 vertebra
2015 Y Liu [46] Rat female SD	DEX (2 mg/kg twice per week for 12 weeks, IM)	Femur and L5 vertebra	Tibia	Femur
2013 Z Ren [50] Rat male SD	DEX (0.1 mg/kg daily for 5 weeks, SC)	Femur	Femur	Femur
2013 F-S Wang [51] Rat male SD	MP (5 mg/kg daily for 1–2 or 4 weeks, SC)	Femur	Femur	Yes at femur (Load (N))
2011 F-S Wang [52] Rat male SD	DEX(0.1 mg/kg daily for 1–2 or 5 weeks, SC)	Bone site no reported	Tibia	Bone site no reported
2012 L Cui [53] Rat male SD	Prednisone (3.5 mg/kg daily for 12 weeks, P.O)	Femur	Tibia	Femur
2017 Y Yang [54] Rat male SD	DEX (1 mg/kg twice per week for 8 weeks, IM)	Femur	Femur	Femur
2014 S Lin [31] Rat male SD	Prednisone (1.5 or 3 or 6 mg/kg daily for 90 days, P.O)	Femur	Tibia at 6 mg/kg and femur at 3 mg/kg	Femur and L5 vertebra at 6 mg/kg
2012 J-Y Ko [55] Rat male SD	DEX (0.1 mg/kg daily for 1–2 or 5 weeks	Bone site no reported	Bone site no reported	Tibia
2020 D Sato [57] Rat female LEW CrlCrlj	Prednisolone (0.42 mg daily for 6 weeks, PSC)	Femur	Tibia	Tibia
2016 A Y Sato [64] Mice female C57BL/6	Prednisolone (1.4 or 2.1 mg/kg daily for 90 days, PSC)	Whole body and L1–L6 vertebrae	L1–L3 vertebrae	L6 vertebra
2019 Q Geng [66] Mice male C57BL/6	DEX (1 mg/kg for 5 days per week for 4 weeks, SC)	Total body, vertebra and femur	Femur	Femur
2018 I Alam [74] Mice male and female Col2.3hWNT16TG	Prednisolone (2.1 mg/kg daily for 28 days, PSC)	Femur	Femur	Femur
2011 M Marenzana [77] Mice BALB/c	DEX (3 mg/kg daily for 6 or 9 weeks, P.O)	Femur	Femur and L5 vertebra	Femur
2014 Z Yongtao [26] Rabbit male New Zealand white	DEX (3 mg/kg twice per week for 12 weeks, IM)	L3–L4 vertebrae	L3 vertebra	L4 vertebra

## Data Availability

The data presented in this study are available on request from the corresponding author.

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
