# Peer review of "Animal Model for Glucocorticoid Induced Osteoporosis: A Systematic Review from 2011 to 2021"

_ijms, 2021, doi:10.3390/ijms23010377_

Round 1

Reviewer 1 Report

In this review the authors provide an extensive and complete overview of animal model for glucocorticoid induced osteoporosis. The manuscript is well written and I have no major points to raise. 

Figure one is in need of a spell-check.

Author Response

We thank  the reviewers for their contributions. The manuscript has been revised according to the comments. Below our reply point by point to the reviewers comments (in red).

Along with our response to the reviewer’s comments, herein, we provide a corrected version with the corrections highlighted and the same version with no track changes.

To the authors

In this review the authors provide an extensive and complete overview of animal model for glucocorticoid induced osteoporosis. The manuscript is well written and I have no major points to raise. 

Response: Thank you we agree and appreciate your comments.

Figure one is in need of a spell-check

Response: The figure one has been corrected as suggested by the reviewer, see in the text page 5.

Thank you again for your time, efforts, and consideration of our work.

Yours Sincerely,

On behalf of the co-authors

Eric Lespessailles

Regional Hospital of Orleans, 14 avenue de l’hôpital, CS 86709,

45067 Orléans Cedex 2, France

Reviewer 2 Report

Xavier et al. aimed to provide an overview of animal models of glucocorticoid-induced bone loss and to show how these models could be useful for preclinical and translational research on GIOP. The proposed systematic review also aims to assess the quality of animal models by focusing on study design, drug dose, timing and size of experimental groups, allocation concealment and outcome measures.

The study covers some issues that have been overlooked in other similar topics. The structure of the manuscript appears adequate and well divided in the sections. Moreover, the study is easy to follow, but some issues should be improved. The manuscript needs some grammar correction. Please also check typos thorough the text.

Conclusion Section: This paragraph required a general revision to eliminate redundant sentences and to add some "take-home message".

Author Response

We thank the reviewers for their contributions. The manuscript has been revised according to the comments. Below our reply point by point to the reviewers comments (in red).

Along with our response to the reviewer’s comments, herein, we provide a corrected version with the corrections highlighted and the same version with no track changes.

Reviewer: 2

To the authors

Xavier et al. aimed to provide an overview of animal models of glucocorticoid-induced bone loss and to show how these models could be useful for preclinical and translational research on GIOP. The proposed systematic review also aims to assess the quality of animal models by focusing on study design, drug dose, timing and size of experimental groups, allocation concealment and outcome measures. The study covers some issues that have been overlooked in other similar topics. The structure of the manuscript appears adequate and well divided in the sections. Moreover, the study is easy to follow, but some issues should be improved.

Response: Thank you for your constructive comments and relevant contribution to improve our manuscript.

The manuscript needs some grammar correction. Please also check typos thorough the text.

Response: We have check typos thorough the text, see in red pages 4,5,10 and 20. All long text, we have done some grammar correction in red.

Conclusion Section: This paragraph required a general revision to eliminate redundant sentences and to add some "take-home message".

Response: We have eliminated redundant sentences and provided a new sentence at the end of the conclusion section as a “take-home message”, see page 20.

Thank you again for your time, efforts, and consideration of our work.

Yours Sincerely,

On behalf of the co-authors

Eric Lespessailles

Regional Hospital of Orleans, 14 avenue de l’hôpital, CS 86709,

45067 Orléans Cedex 2, France 
